# Unobtrusive Continuous Stress Detection in Knowledge Work—Statistical Analysis on User Acceptance

**Johanna Kallio** [1,*] , **Elena Vildjiounaite** [1] , **Julia Kantorovitch** [2] , **Atte Kinnula** [1] **and Miguel Bordallo López** [1,3]

[1]  Knowledge Intensive Products and Services, VTT Technical Research Centre of Finland Ltd., Kaitoväylä 1, FI-90571 Oulu, Finland; Elena.Vildjiounaite@vtt.fi (E.V.); Atte.Kinnula@vtt.fi (A.K.); miguel.bordallo@vtt.fi or miguel.bordallo@oulu.fi (M.B.L.)

[2]  Data Intensive Economy, VTT Technical Research Centre of Finland Ltd., Tekniikantie 21, FI-02150 Espoo, Finland; Julia.Kantorovitch@vtt.fi

[3]  Faculty of Information Technology and Electrical Engineering, University of Oulu, Pentti Kaiteran katu 1, FI-90570 Oulu, Finland

[*]  Correspondence: Johanna.Kallio@vtt.fi

**Abstract:** Modern knowledge work is highly intense and demanding, exposing workers to long-term psychosocial stress. In order to address the problem, stress detection technologies have been developed, enabling the continuous assessment of personal stress based on multimodal sensor data. However, stakeholders lack insights into how employees perceive different monitoring technologies and whether they are willing to share stress-indicative data in order to sustain well-being at the individual, team, and organizational levels in the knowledge work context. To fill this research gap, we developed a theoretical model for knowledge workers' interest in sharing their stress-indicative data collected with unobtrusive sensors and examined it empirically using structural equation modeling (SEM) with a survey of 181 European knowledge workers. The results did not show statistically significant privacy concerns regarding environmental sensors such as air quality, sound level, and motion sensors. On the other hand, concerns about more privacy-sensitive methods such as tracking personal device usage patterns did not prevent user acceptance nor intent to share data. Overall, knowledge workers were highly interested in employing stress monitoring technologies to measure their stress levels and receive information about their personal well-being. The results validate the willingness to accept the unobtrusive, continuous stress detection in the context of knowledge work.

**Keywords:** stress; unobtrusive detection; data sharing; human–computer interaction; human factors; modeling; survey

## 1. Introduction

The amount of knowledge-intensive work has been continuously increasing in modern societies; up to 50% of workers are knowledge workers [1]. Knowledge workers can be defined as skilled and autonomous workers who create and apply knowledge in order to produce complex results [2]; however, their work is highly intensive, cognitive, and emotionally demanding, making them prone to psychosocial stress and fatigue [3]. The International Labor Organization (ILO) defines work-related stress as "the harmful physical and emotional response caused by an imbalance between the perceived demands and the perceived resources and abilities of individuals to cope with those demands" [4] (p. 2). Long-term stress is associated with various mental health problems (anxiety, insomnia, depression, fatigue, and concentration difficulties), cardiovascular diseases, poor immune function, and presenteeism [5]. The cost of work-related stress is hundreds of billions of euros annually worldwide [3].

The United Nations has defined two relevant sustainable development goals: (a) to ensure healthy lives and promote well-being for all at all ages, and (b) to promote sustained,

inclusive and sustainable economic growth, full and productive employment, and decent work for all [6]. Correspondingly, employee well-being has emerged as a strategic priority in all organizations facing ongoing demographic and technological changes [7]. Healthy, skilled, and motivated employees are seen as the most important capital of knowledge-intensive organizations in the increasingly fierce global competition and pace of work, and with extended working life [4,8]. To maintain sustainable work, organizations invest in various physical and mental programs, but such programs lack the capabilities to support employees individually and measure the impact of well-being programs [9]. Instead of periodic surveys, organizations are looking for well-accepted solutions to continuously measure employee well-being, to minimize health risks, and to avoid adverse outcomes [7]. Novel sensor-based well-being technologies facilitate the accurate stress assessment of employees in real time [10], but personal well-being data are more privacy sensitive than many other types of data, and thus, continuous stress detection requires user acceptance before it can be applied to well-being at work.

This study aims to provide insights into how knowledge workers perceive different automatic stress-detection technologies and personal data sharing in the work context. The focus was specifically on measuring the employees via unobtrusive sensors embedded in the work environment (either discreetly or virtually), thereby enabling the passive extraction of data and freeing the employees to conduct their usual daily activities without disturbance from the measurement system itself (e.g., charging, wearing, loss of privacy). In order to deploy the continuous stress monitoring concept for broader adoption, it is essential to answer the following questions: (1) Will employees accept stress detection? (2) What is the overall level of interest in using the resulting well-being information? (3) Are there technologies that are more privacy-sensitive than others? and (4) How interested are the users to share personal data in exchange for added benefits? Most well-being monitoring acceptance studies in the work context have investigated the employees' intentions to adopt wearable technologies [11–13]. However, to the best of our knowledge, there have not been any surveys concerning the knowledge workers' perceptions of different stress monitoring technologies and their interest in sharing personal stress-indicative data in the work context in order to sustain well-being at the individual, team, and organizational levels.

The main contribution of this research is to provide a statistically valid and conclusive answer to the research questions above-mentioned in order to determine the feasibility of continuous stress monitoring in the knowledge work context. To this end, we developed a theoretical model for the willingness of knowledge workers to share data collected with unobtrusive sensors to sustain well-being and tested it empirically with structural equation modeling (SEM) and anonymous online survey data. Moreover, the perceptions of knowledge workers of different stress-detection technologies and their privacy sensitivity were studied via the conducted survey. The quantitative survey was conducted in spring 2020 and resulted in 181 responses from European knowledge-intensive organizations. Overall, the results validate the knowledge workers' willingness to accept the unobtrusive, continuous stress detection and data sharing in order to promote well-being at work.

### 1.1. Unobtrusive Stress Detection

Stress is manifested in psychological, physiological, and behavioral responses in everyday life. Psychological responses are related to emotions and mental processes, whereas physiological responses refer to the activation of hypothalamic–pituitary–adrenal systems and autonomic nervous systems such as increased heart rate, respiration, and sweating [14]. Behavioral responses include, for instance, motion, postures, facial expressions, and the usage of digital devices, reflecting an individual's emotions and cognition [15].

At present, work-related stress is typically assessed using either periodic or occasional surveys, mainly focusing on the employees' perceived physiological responses such as emotions or mood (e.g., [16]). However, the major limitations of these surveys are their infrequency, the low number of measurements, and the high data collection and analysis effort. More frequent but less burdensome solutions to measuring work-related stress fac-

tors and supporting employee well-being promptly are required. Therefore, stress research using sensing, computing, and communications technologies suitable for continuous stress monitoring has become an active research area [10].

Wearable devices comprising physiological sensors have demonstrated a high potential for objective stress detection in laboratory studies [10], but disturbance induced by wearing and charging, and privacy risks affect the adoption of wearables in long-term use [17,18]. Moreover, when measurement devices are worn or otherwise attached to a person, real-world measurements typically produce imperfect output data [19]. To overcome the discomfort and data-loss challenges of wearable devices, continuous stress monitoring based on unobtrusive sensors and software (i.e., virtual sensors without a physical form) embedded in the work environment have been proposed as an additional data source or the only data source for long-term use [17,20].

Previous studies have collected behavioral data from computer, keystroke, and mouse dynamics such as interaction time, typing pressure, and mouse clicking, but mostly under laboratory conditions [21–25]. The majority of real-life stress-detection studies have been based on behavioral data gathered from smartphones [26–32]. The behavioral data collection approaches are convenient because they do not require any additional gadgets, but using data from personal devices can pose privacy concerns. However, when monitoring is focused on the usage patterns (e.g., writing tempo, motion velocity, program categories, duration) instead of content (e.g., what is written, what the user clicks on, what programs or applications are used), the methods have been evaluated as being notably less privacy threatening [10,33].

Computer vision is another well-studied behavioral stress-detection approach that focuses on analyzing facial expressions, postures, and eye movements [10,33,34]. Video cameras can be deployed to assess stress and emotional states relatively accurately in laboratory conditions; however, their major disadvantage is the lack of privacy, making them an undesirable and obtrusive option from a user perspective [33]. On the other hand, a depth sensor is a more privacy-safe type of computer vision for the reason that the monitored person is not readily identifiable from the depth image data. Moreover, the depth sensor can be positioned in office ceilings to have a side-view of the monitored persons and detect head trajectories (i.e., head motion) instead of pointing directly at faces [35].

In addition, earlier studies have recognized smart work environments as an option to collect behavioral data from employees [10]. Motion or postures extracted from passive infrared sensors and pressure-sensitive chair data can provide information about the workers' stress levels [35,36]. Interestingly, data from low-cost in-office sensors such as environmental quality sensors installed in the walls or ceilings can also be indicative of stress levels [37]. Compared with smartphone-, computer-, or keyboard-usage-based stress detection, these environmental sensors can be considered less privacy threatening.

Previous studies have suggested that privacy perspectives are critical factors in adopting wearable devices [18], while cost-efficiency and easy deployment are essential in real-life stress detection implementations [33]. Thus, this study focused on unobtrusive stress monitoring methods embedded in the work environment that do not require user effort (e.g., wearing, charging) or attention (e.g., interfering with performance, privacy threat), and that can be applied in knowledge work over the long-term.

### 1.2. The Theoretical Context of Behavioral Intentions

Behavioral human sciences have introduced several decision-making theories in order to conceptualize and understand social behaviors, but predicting human intentions and actions (i.e., willingness to use technology and share data) is challenging, especially at an organizational level. Sociotechnical systems theory has approached organizational excellence and well-being by studying the interaction between humans and technology (or the environment) in workplaces [38]. According to Eason [38], sociotechnical systems can be defined as heterogeneous, consisting of social and technical components with divergent



characteristics. Compared to technical components, humans are constantly aware of their environment and are capable of changing their behavior. Therefore, considering user perspectives is an integral part of technology development that includes human–technology interaction.

Structural decision-making models such as the theory of planned behavior (TPB) proposed by Ajzen [39] can be considered as a well-established basis for predicting an individual's behavior. The TPB suggests that an individual's behavior depends on the person's intention to perform an action, indicated by the individual's attitude, perception of subjective norms, and perceived behavioral control [39]. Earlier studies have successfully applied the TPB to predict data sharing intentions in different contexts including scientific research [40]. Following [39], the attitude is linked with an individual's perceptions about the possible outcomes of sharing data, and the subjective norm is a person's belief about other people's expectations toward an action (i.e., data sharing). The perceived behavioral control is related to the individual's perceptions of their personal ability to perform a given action.

The technology acceptance model (TAM) [41] and the unified theory of acceptance and use of technology (UTAUT) model [42,43] extend the TPB and provide explanations for user acceptance and the usage of technology. TAM explains a person's intention to use technology as depending on their perceptions of usefulness and ease of use in regard to the technology in question. TAM and the UTAUT have been widely implemented and tested in wearable technology-related studies in diverse work contexts (e.g., [12,14,44]). Williams et al. [43] found performance expectancy in the UTAUT model to be the most vital factor in predicting technology's actual use, meaning that people are more willing to use technology such as sensors and tracking software when they believe that the technology helps them.

Some technology acceptance models have been extended with concepts of trust and intrusiveness including privacy aspects (e.g., [13,18]). A theoretical basis for privacy concerns can be applied when considering continuous sensor-based stress monitoring and sharing personal data that is indicative of the employees' stress levels. Two antecedents can define the perceived privacy concerns, namely perceived vulnerability and perceived ability to control information [45]. These two factors influence the employees' privacy concerns when they decide whether to share the data collected on themselves. When individuals perceive that their data will not be used fairly and/or that there will be negative consequences, they will be less likely to employ sensors or tracking software while working [11,46]. In other words, individuals with serious concerns regarding their data's misuse will seek to minimize their vulnerability by refusing data sharing to promote their well-being.

In summary, a person's behavior is linked to the person's motivation and intentions. More positive attitudes, more substantial positive social pressure, and greater perceived control will lead to stronger motivation and intentions. Therefore, we can assume that a positive attitude toward unobtrusive stress monitoring technology will positively influence their intentions to share data in order to support well-being. On the other hand, deploying sensor-based stress detection and sharing personal data requires trust, security, and privacy protection, especially in the work context. Although the TPB, TAM, and the UTAUT have successfully been applied in data sharing and well-being technology in diverse contexts, they have also been criticized for direct intention–behavior linkage and limited prediction value [47]. Thus, this study does not aim to test these models, but uses them as a basis for developing a general model for sharing personal well-being data collected with unobtrusive sensors in knowledge-intensive work.

The remaining part of this paper is structured as follows. Section 2 introduces the research hypothesis and describes the employed methods and empirical survey data. The empirical data analysis results are presented in Section 3, and the theoretical and empirical contributions of the study are discussed in Section 4. Finally, we summarize our conclusions and provide an outlook for future work in Section 5.

## 2. Materials and Methods

### 2.1. Research Model and Hypotheses

Based on the TPB, TAM, and theory of privacy concerns, a research model was developed to predict the intentions of knowledge workers to share their stress-indicative data in order to sustain well-being at individual, team, and organizational levels. The model determines the employees' intention to share personal data in the work context by assessing their interest in employing unobtrusive stress monitoring technologies and their privacy concerns about data use (see Figure 1). The concepts of the research model were "Interest in employing environmental sensors" (concept code ENV), "Interest in employing tracking software" (concept code TRA), "Privacy concerns about the use of data" (concept code CON), and "Interest in sharing personal data" (concept code SHA). The TPB explains the motivation behind the employees' data sharing intentions, and TAM establishes their intent to use novel stress detection technology (i.e., their motivational belief that sharing data and using technology will help them). Following this, it can also be assumed that a positive attitude toward stress detection technology is linked with an interest in sharing personal data in order to sustain well-being. Moreover, the theory of privacy concerns explains the underlying privacy factors affecting the employees' willingness to employ stress monitoring and share personal data in the work context. Thus, the main hypotheses of this study are as follows:

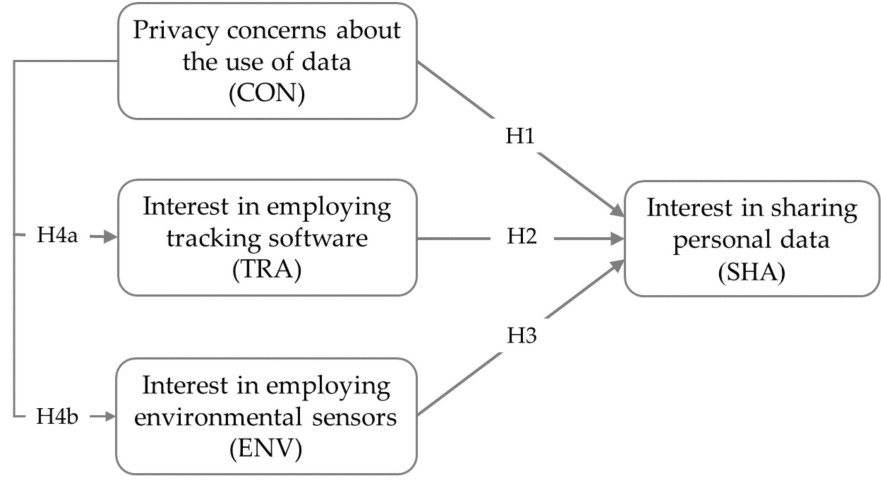

**Figure 1.** The concepts and hypotheses of the study.

**Hypothesis 1 (H1).** *Employee privacy concerns about using data negatively influence their interest in sharing stress-indicative personal data in order to sustain their well-being.*

**Hypothesis 2 (H2).** *Employee interest in employing tracking software to monitor their stress positively influences their interest in sharing personal data in order to sustain their well-being.*

**Hypothesis 3 (H3).** *Employee interest in employing environmental sensors for monitoring their stress positively influences their interest in sharing personal data in order to sustain their well-being.*

**Hypothesis 4 (H4).** *Employee privacy concerns about using data negatively influence their interest in employing tracking software (a) and environmental sensors (b) in order to monitor their stress.*

Figure 1 depicts the research model for knowledge workers' interest in sharing data.

### 2.2. Empirical Data Collection

The empirical data about knowledge workers' perceptions regarding sensor-based stress monitoring and well-being solutions in the work environment were collected anony-

mously in spring 2020 using the Internet and the Questback Inc. online survey tool. The survey was conducted by the VTT Technical Research Center of Finland Ltd., and the survey link was distributed on their online news page, and Twitter and LinkedIn channels, followed by a large number of knowledge workers. The idea was to reach a wide range of European knowledge workers; however, the distribution choice did not allow counting the response rate.

This study's structural model is a second-order model. Hence, the statements in the survey questionnaire measured theoretical concepts (see Table 1). The researchers designed the questions related to the employees' interest in sharing personal data, interest in using sensors and tracking software, and concerns about the use of data in the knowledge work context while considering the main features of the concepts in the research model based on existing theories. Continuous stress monitoring employing behavioral data from unobtrusive sensors and trackers is a relatively new research topic. Thus, operational measures (i.e., questions) were not directly available from the earlier research, although previous studies were used as a basis for them (e.g., [11]). The questions were measured on a four-point Likert scale (1 = not at all interested, 2 = not that interested, 3 = interested, 4 = very interested for Q1, 1 = strongly disagree, 2 = disagree, 3 = agree, 4 = strongly agree for Q3–Q6 and 1 = not at all concerned, 2 = not that concerned, 3 = concerned, 4 = very concerned for Q7) and included attitudinal statements.

**Table 1.** Constructs, their operational measures, and descriptive statistics.

| Construct | Operational Measure in the Questionnaire | Mean | SD |
|---|---|---|---|
| | Q1. How interested would you be in receiving information about your well-being, measured with sensors during the working day, if the information is only in your use? | 3.12 | 0.78 |
| | Q2. What kind of measurable information related to your well-being and coping at work would you be interested in receiving? | - | - |
| | Q3. How interested would you be to employ a sports watch or other wearables in order to monitor your stress level during the workday if the data are only in your use? | 3.46 | 0.62 |
| ENV | Q4. How interested would you be to employ the following sensors embedded in the work environment in order to monitor your stress level during the workday if the data are only in your use? | | |
| E1 | Air quality sensors | 3.47 | 0.66 |
| E2 | Sound level sensors | 3.30 | 0.81 |
| E3 | Motion detectors | 3.05 | 0.81 |
| E4 | A pressure-sensitive chair | 3.20 | 0.74 |
| | A video camera [1] | 1.95 | 0.85 |
| TRA | Q5. How interested would you be to employ software that tracks your way of using the following device to monitor your stress level during the workday if the data are only in your use? | | |
| T1 | A keyboard- or mouse-usage tracker | 3.15 | 0.89 |
| T2 | A computer-usage tracker | 2.98 | 0.93 |
| T3 | A smartphone-usage tracker | 2.87 | 0.95 |
| SHA | Q6. I would be interested in confidentially sharing data collected from myself during the workdays with related well-being service providers: | | |
| S1 | if the data would be used to help in identifying my personal health risks. | 3.15 | 0.88 |
| S2 | if the data could be used to improve my own and my colleagues' well-being and coping at work. | 2.96 | 0.85 |
| S3 | if the data could be used to improve my organization's work culture and leadership in a direction that supports well-being and coping at work. | 2.88 | 0.96 |
| S4 | if the data would only be used for non-commercial scientific purposes. | 3.13 | 0.87 |

**Table 1.** *Cont.*

| Construct | Operational Measure in the Questionnaire | Mean | SD |
|---|---|---|---|
| CON | Q7. If your stress was monitored while you were working, how concerned would you be about the following things? | | |
| C1 | My employer, supervisor, or co-workers could use the data collected against me. | 2.74 | 0.86 |
| C2 | My employer could get private or sensitive information about me. | 3.04 | 0.81 |
| C3 | Someone who is not supposed to see my data could get access to my personal data. | 3.22 | 0.75 |
| | Q8. What do you consider to be the most privacy-sensitive stress monitoring methods? | | |

[1] Not applied in the empirical model because video camera was classified as obtrusive.

Moreover, the respondents were asked two multiple-choice questions: Q2 and Q8. The options for Q2 were: stress level, workload, performance, concentration level, recovery, ergonomics, and heart rate/heart rate variability; and for Q8: A sports watch, an air quality sensor, a sound level sensor, a motion detector, a video camera, a pressure-sensitive chair, a keyboard, a mouse-usage tracker, a computer-usage tracker, and smartphone-usage tracker. In addition, the survey contained an open field for comments.

*2.3. Statistical Analysis*

Preliminary statistical analyses were conducted with SPSS Statistics 26 software. For advanced SEM analyses, the Mplus Version 8.4 Base Program was used. SEM is a statistical technique that can test and estimate the reliability and validity of theoretical constructs and their inferential relationships [48]. Thus, we chose SEM for explanatory purposes in this study. The estimates were calculated using the maximum likelihood method, based on a covariance matrix. The empirical modeling inputs were designed and selected based on their feasibility regarding unobtrusiveness (embedded and not noticeable); thus, the question of interest in using video cameras was not applied in the empirical model. In the next section, the proposed constructs and their relationships are empirically tested in the knowledge work context, and the operational measures of theoretical constructs are validated.

**3. Results**

*3.1. Dataset Statistics*

The conducted survey obtained a total of 181 responses from European knowledge workers living in 12 different counties. Table 1 presents all the survey questions and descriptive statistics including the mean and standard deviation (SD) of the four-point Likert scale responses and Table 2 shows the profile of the survey respondents. The survey data are available as a public dataset via Zenodo.

The Kolmogorov–Smirnov significance level was below 0.05 for all observed study variables (i.e., measures), which means that survey data were non-normal. The difference between the responses in each profile group (i.e., gender, age, knowledge field, and nationality) was compared using Kruskal–Wallis analyses. There were no statistically significant differences (using the criterion of $p > 0.05$) between responses, and it seems justifiable to conclude that the sample profile did not have a statistically significant effect on this study's responses.

**Table 2.** Demographics of the survey respondents.

| Profile Category | Percentage |
|---|---|
| **Gender** | |
| Female | 43.6 |
| Male | 56.4 |
| **Age** | |
| 20–24 years old | 1.7 |
| 25–34 years old | 25.4 |
| 35–44 years old | 42.0 |
| 45–54 years old | 20.4 |
| 55 or more years old | 10.5 |
| **Country of residence** | |
| Finland | 63.5 |
| Spain | 26.4 |
| Germany | 2.2 |
| Other (including Canada, Italy, Mexico, the Netherlands, Norway, Luxemburg, Sweden, Switzerland, and the USA) [1] | 7.9 |
| **Knowledge field** | |
| Information and communication technology | 42.9 |
| Engineering, manufacturing, and construction | 28.6 |
| Natural sciences and mathematics | 10.7 |
| Business, administration, and law | 5.4 |
| Health and welfare | 5.4 |
| Education | 2.4 |
| Services | 2.4 |
| Other fields | 2.2 |
| **Work position** | |
| Specialist | 69.1 |
| Manager | 17.1 |
| Entrepreneur, self-employed | 4.4 |
| Assistant | 3.9 |
| Other | 5.5 |

[1] For these counties of residence with less than five responses, only the total percentage is reported due to privacy reasons.

### 3.2. Perceptions toward Different Stress Monitoring Methods

To better understand people's motivations, we asked (Q1) if the respondents were interested in receiving information about their well-being, measured during workdays, and (Q2) what kinds of well-being-related information they would be interested in receiving. Answers to Q1 revealed that the majority (81.8% ± 5.6% at 95% confidence level) of the respondents ($n$ = 181) were very interested or interested in receiving well-being related information. Less than a fifth of the respondents (18.2% ± 5.6%) stated that they were not that interested or not at all interested in receiving such information.

Regarding Q2, most of the respondents were interested in receiving information regarding their stress (75.1% ± 6.3% of the respondents were interested) or concentration (66.9% of the respondents were interested). Roughly 50 % (±7.3%) of the respondents were willing to know about their performance and ergonomics. Information about recovery/coping and workload were interesting topics for around 45 % (±7.3%) of the respondents. Heart rate or heart rate variability was found to be interesting among 39.8 % (±7.1%) of the respondents. A minority (2.7% ± 2.4%) were interested in another type of well-being information, but they did not specify it in more detail. Figure 2 illustrates the survey responses regarding the interesting well-being information types.

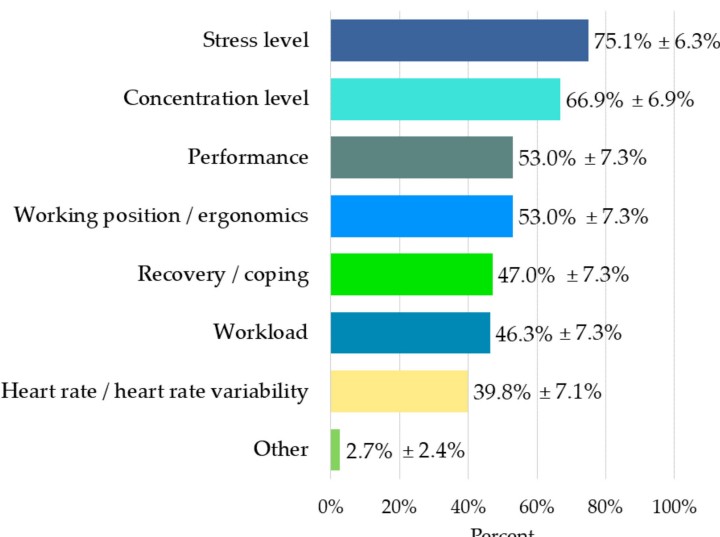

**Figure 2.** Knowledge workers' interest toward different well-being information with a confidence interval at the 95% confidence level.

According to the answers to Q3–Q5, the respondents were most interested in using air quality sensors (mean = 3.47, SD = 0.66) or wearables (mean = 3.46, SD = 0.66) to monitor their stress level during workdays. Sound level sensors, a pressure-sensitive chair, a keyboard- or mouse-usage tracker and motion detectors, a computer-usage tracker, and a smartphone-usage tracker were also considered as interesting options. The respondents were most reluctant to use a video camera (mean = 1.95, SD = 0.85) for stress monitoring. Table 1 presents the descriptive statistics.

We were also eager to explore (using Q8) which stress monitoring methods were the most privacy sensitive (from the knowledge workers' perspective) when used in the work context. A video camera was voted as the most privacy sensitive (i.e., intrusive method), gaining 82.3% ± 5.6% of the votes. Roughly 70 % (±6.7%) of the respondents also considered smartphone-usage and computer-usage trackers as privacy sensitive, whereas about 30 % (±6.8%) rated sports watches and mouse-usage trackers as privacy sensitive. The least sensitive methods were air quality sensors, a pressure-sensitive chair, a sound level sensor, and a motion detector. Figure 3 illustrates the response distribution between the different stress monitoring methods.

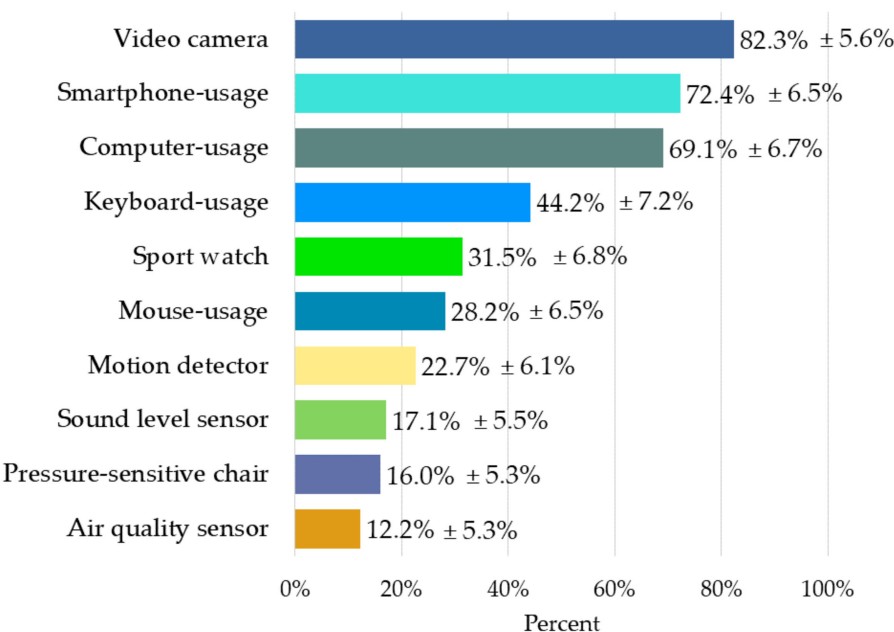

**Figure 3.** Perceptions of the most privacy-sensitive stress monitoring methods with a confidence interval at the 95% confidence level.

### 3.3. Modeling

We tested the research model for the knowledge workers' interest in sharing their stress-indicative data in the work context with SEM. The empirical model projected an almost acceptable statistical fit (see Table 3). A root mean square error of approximation (RMSEA) value below 0.08 represents the good fit of the model [49], and Tucker–Lewis index (TLI) and comparative fit index (CFI) values above 0.90 indicate an acceptable fit between the hypothesized model and the observed data [50]. Moreover, a standardized root mean squared residual (SRMR) value below 0.08 supports the model's good statistical fit [51]. However, the $p$-value of the $\chi^2$ test might not support the model fit, and the model contains some statistically insignificant constructs (see the Appendix A, Figure A1, for the empirical model). Furthermore, construct reliability (CR) should be 0.70 or greater, and the reliability value for average variance extraction (AVE) should be 0.50 or greater [52]. Thus, the empirical model was not valid as such, and we continued the analysis.

**Table 3.** The test statistics of the initial empirical model.

| Initial Empirical Model's Fit | | | Construct Reliabilities for Latent Variables | | | |
|---|---|---|---|---|---|---|
| Chi-square | 129.63 (df. 71, $p$ = 0.000) | | SHA | CON | TRA | ENV |
| RMSEA | 0.066 | AVE | 0.64 | 0.28 | 0.42 | 0.45 |
| CFI | 0.928 | CR | 0.86 | 0.62 | 0.77 | 0.75 |
| TLI | 0.908 | Cronbach's alpha | 0.87 | 0.77 | 0.89 | 0.76 |
| SRMR | 0.062 | | | | | |

To further study the concepts, relationships, and constructs, we carried out the empirical analysis in an exploratory manner with the Mplus software. We produced an amended model, which is illustrated in Figure 4. In this model, the concern about the use of data demonstrates a negative relationship with the interest to share personal data (H1) and the interest to employ tracking software (H4a), as assumed in the theoretical model. However, there was no relationship between the concern about the use of data and the interest to

employ environmental sensors for stress monitoring (H4b), which was not considered in the theoretical model.

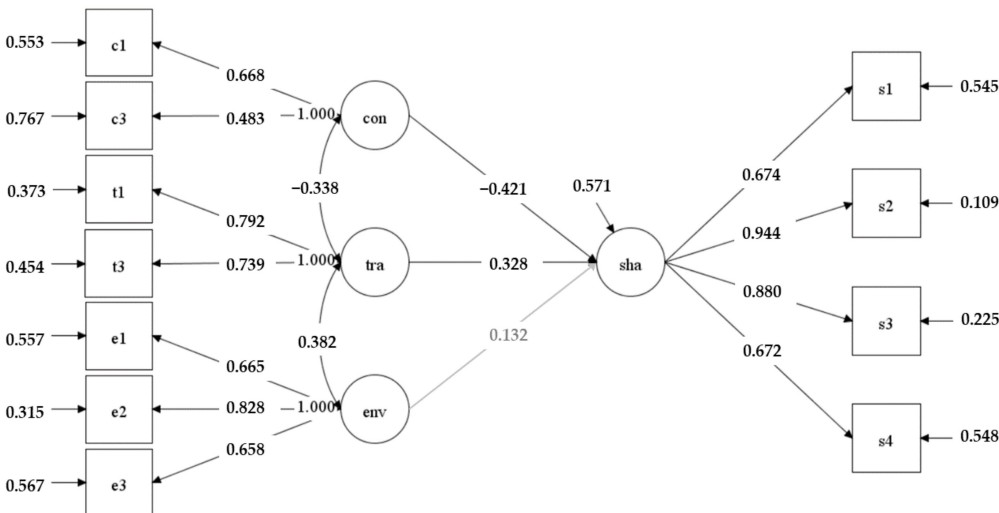

**Figure 4.** The amended empirical model.

The interest in employing tracking software showed a positive relationship with the interest in sharing personal data to improve well-being at the personal, team, and organizational levels (H2). In addition, the empirical data showed a positive relationship between the interest in employing environmental sensors and the interest in employing tracking software. Against our expectation, the interest in employing environmental sensors did not indicate a statistically significant relationship with the interest in sharing personal data (H3). However, the interest in employing environmental sensors had an indirect positive effect on the interest to share personal data, and that effect was mediated by the interest in employing tracking software.

The test statistics of the empirical model are presented in Table 4. The $\chi^2$ test showed an acceptable fit of the model, the *p*-value being 0.04. Similarly, RMSEA below 0.08, CFI and TLI above 0.90, and SRMR below 0.08 indicated a good fit for the model. Thus, based on these test values, the amended model was acceptable. Moreover, we evaluated each latent variable individually (see Table 4). The validity of the measures can be extrapolated from factor loadings [53], which are acceptable for the latent variables SHA, TRA, and ENV. However, in the amended model, CON had only two measures impairing the usability of its CR and AVE values. The weakness of these values can be explained by the fact that CON did not have the suggested three or more measures as well as by the somewhat high error terms. Nonetheless, a Cronbach's alpha of 0.7 or higher supports the validity of CON's measures.

**Table 4.** The test statistics of the amended empirical model.

| Amended Empirical Model's Fit | | | Construct Reliabilities for Latent Variables | | | |
|---|---|---|---|---|---|---|
| Chi-square | 55.558 (df. 39, *p* = 0.04) | | SHA | CON | TRA | ENV |
| RMSEA | 0.048 | AVE | 0.64 | 0.33 | 0.59 | 0.52 |
| CFI | 0.965 | CR | 0.87 | 0.50 | 0.78 | 0.76 |
| TLI | 0.950 | Cronbach's alpha | 0.87 | 0.84 | 0.74 | 0.76 |
| SRMR | 0.057 | | | | | |

## 4. Discussion

Managing stress is a critical aspect of the sustainable work–life well-being of individual employees, organizations, and society. Despite the amount of research on stress detection based on multimodal sensor data (e.g., [10,33]), the employees' perceptions of employing unobtrusive stress monitoring and sharing personal data for well-being improvements in the work context are still not well established in the engineering discipline. The purpose of this study was to promote a sustainable work culture by adopting novel sensing technologies. In particular, the deployment of a data-driven approach to assessing stress and building up a supportive work environment calls for employee acceptance.

As a response to whether people would accept stress detection, the survey results revealed that the knowledge workers were highly interested in using unobtrusive stress monitoring technologies to monitor their stress levels during workdays. The different stress monitoring methods including environmental sensors, tracking software, and more obtrusive wearables were all ranked as attractive options. The high interest in wearables may be due to respondents having previous experience with wearables [11]. However, wearables require usage diligence from the user (e.g., charging and wearing) and some users consider them obtrusive; hence, a stress detection approach using sensors embedded in the environment may work better in the long-term. For instance, some respondents commented that "using in-office sensors for stress monitoring is ok because sensors do not require actions from the user." Moreover, the overall level of interest in using the resulting well-being information was high ($81.8\% \pm 5.6\%$); the most interesting topics were the stress level ($75.1\% \pm 5.6\%$ of the respondents) and concentration level ($66.9\% \pm 6.9\%$ of the respondents).

Regarding the question about the monitoring technologies' privacy sensitivity, the respondents were most unwilling to use video camera data for stress monitoring, which is congruent with the evaluation results of Carneiro et al. [33]. For instance, several respondents commented that they were suspicious and felt insecure about using video camera data. Moreover, video cameras, smartphones, and computer-usage data-based monitoring methods were considered privacy threatening. This ranking is understandable because video stream, smartphone, and computer-usage data may expose the employees' private and sensitive information, causing negative consequences. Instead, using environmental sensors such as air quality, sound level, and motion sensors may overcome privacy issues.

Concerning sharing personal stress information for the exchange to added benefits, we developed a theoretical model and tested it empirically. Although scholars have studied technology acceptance and employees' willingness to use wearables at work (e.g., [11,13,44]), to the best of our knowledge, no one has investigated the interest of knowledge workers in sharing behavioral data that is indicative of stress in order to sustain well-being at work. Therefore, providing a workable second-order construct of the knowledge workers' interest in sharing personal data in order to sustain well-being advances the literature on the behavioral intentions related to data sharing in work, well-being, and health contexts. The empirical analysis resulted in a statistically valid amended model, which is depicted in Figure 4. In addition, we validated the operational measures of the theoretical constructs. Based on the empirical model, people seemed most willing to share personal stress data when applied for health research and personal interventions. Those who had a more positive attitude toward employing usage pattern-based stress detection were also more willing to share their stress-indicative data.

Overall, the results confirmed the feasibility of a continuous and unobtrusive stress monitoring approach in a knowledge work context. Interestingly, based on the analysis results, privacy concerns did not affect the knowledge workers' interest in using environmental sensors such as air quality, motion, and sound level sensors, indicating that environmental sensors are less intrusive and thus more acceptable when assessing work-related stress. This finding is significant and can partly solve the privacy issues related to continuous stress monitoring and personal data sharing. On the other hand, privacy concerns related to employing more privacy-sensitive methods such as tracking software

for stress monitoring did not prevent data sharing intentions. However, our study confirms that privacy is an essential element when using employees' well-being data in the work context, as pointed out earlier (e.g., [18]).

The General Data Protection Regulation (GDPR) governs the rights of EU citizens to data protection and personal-data processing confidentiality. Since the invocation of sensors and cloud-based data analytics is growing, it is crucial to investigate the GDPR directive regarding the design and implementation of data-driven solutions in order to minimize conflicts between GDPR and the deployment of sensors and tracking software-based well-being monitoring [54]. Nevertheless, despite privacy concerns, particularly regarding keyboard-, mouse- and smartphone-usage data, people are still willing to adopt the proposed technologies. This would imply that the associated benefit is considered significant enough to override the concerns.

Although this study provided theoretical and practical contributions, there are some limitations. The empirical data consisted of 181 responses among European knowledge-intensive organizations, where 89.9 % of the respondents were from Finland and Spain and only 10.1 % were from other countries. Thus, future work should include further empirical model coverage and a broader nationality scope, although gender, age, and nationality did not influence the observed variables in this study. Moreover, the resultant empirical model that examined the causal relations was a simplification of a specific view of the world and could be extended to cover other influencing factors at individual and organizational levels. Nevertheless, even in a simple form, the resultant model provides useful information for work-related well-being service designers to adopt.

## 5. Conclusions

The workforce's mental health has become an increasing problem worldwide because of stress, depression, and anxiety. Particularly in knowledge-intensive work, employees' mental health has emerged as a strategic priority in order to maintain workforce sustainability. Thus, new solutions that aim to mitigate stress-related risk factors and sustain work well-being at individual, team, and organizational levels are necessary. Accordingly, the goal of this study was to provide insights into how knowledge workers perceive different stress-detection technologies and personal data sharing in the work context.

This study has some considerable contributions to highlight. Theoretically, the study extends the related literature by proving a second-order construct of the knowledge workers' interest in sharing personal data in order to improve well-being in the work context. The model was tested empirically using SEM and data collected from European knowledge-intensive organizations, which produced the statically valid amended model. Moreover, the operational measures for the theoretical constructs were validated. The quantitative analysis results revealed that privacy concerns did not apply to the willingness to use environmental sensors such as air quality, sound level, and motion sensors. On the other hand, the concerns about more privacy-sensitive stress detection methods did not prevent user acceptance nor intent to share data.

Moreover, the study showed that knowledge workers in Europe are eager to receive well-being-related information, especially regarding their stress and concentration levels. The respondents were also highly interested in using stress monitoring technologies such as environmental sensors, pressure-sensitive chairs, and keyboard- or mouse-usage trackers. Therefore, our study confirmed that employing a continuous and unobtrusive stress monitoring approach in a knowledge work context is feasible.

The future goal of personal-stress data sharing in a workplace context is to develop a secure and privacy-safe organizational barometer that aggregates employees' data anonymously and provides employees with the means to make their discomfort visible to managers. In line with the United Nations' Sustainable Development Goals, the ultimate aim is to better support employee health and facilitate both an empathic workplace culture and psychological safety in demanding knowledge-intensive work.

**Author Contributions:** Conceptualization, J.K. (Johanna Kallio), E.V., J.K. (Julia Kantorovitch), A.K., and M.B.L.; Methodology, J.K. (Johanna Kallio); Formal analysis, J.K. (Johanna Kallio); Writing—original draft, J.K. (Johanna Kallio); Writing—review and editing, J.K. (Johanna Kallio), E.V., J.K. (Julia Kantorovitch), A.K., and M.B.L.; Visualization, J.K. (Johanna Kallio); Supervision, M.B.L.; Project administration, A.K.; Funding acquisition, J.K. (Johanna Kallio), E.V., J.K. (Julia Kantorovitch), and A.K. All authors have read and agreed to the published version of the manuscript.

**Funding:** This research was funded by Business Finland under ITEA 18033 Mad@Work, grant number 2991/31/2019.

**Institutional Review Board Statement:** The study was conducted according to the guidelines of the Declaration of Helsinki, and approved by the Institutional Review Board of VTT Technical Research Center of Finland Ltd. (date of approval 5 December 2019).

**Informed Consent Statement:** Informed consent was obtained from all subjects involved in the study.

**Data Availability Statement:** The data presented in this study are openly available via Zenodo at doi: 10.5281/zenodo.4457393.

**Acknowledgments:** We are thankful to all the survey respondents for their significant role in the study. We are grateful to Mari Juntunen and Jouni Juntunen for their comprehensive guidance on the theoretical background and quantitative analysis methods. We acknowledge our partners in collaboration and colleagues for sharing the questionnaire link in their networks. Aside from the above, we wish to thank Satu-Marja Mäkelä, Juha Häikiö, Panu Korpipää, and Marianne Kinnula for their helpful discussions.

**Conflicts of Interest:** The authors declare no conflict of interest.

## Appendix A

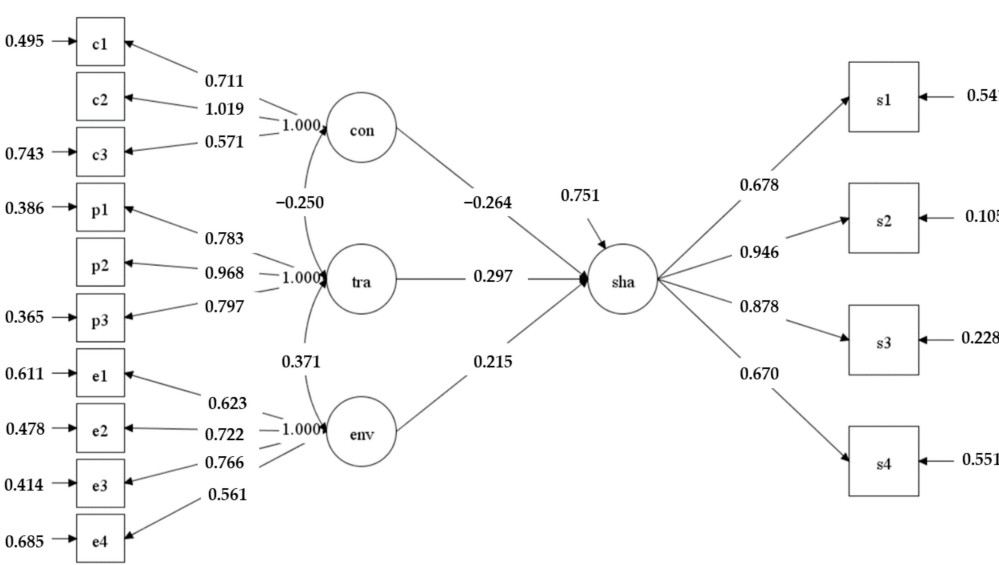

**Figure A1.** The initial empirical model.

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
