# Peer review of "Unobtrusive Continuous Stress Detection in Knowledge Work—Statistical Analysis on User Acceptance"

_sustainability, doi:10.3390/su13042003_

Round 1
Reviewer 1 Report
Dear Authors,
Thank you for your submission. Your manuscript is interesting, sound and well written.
I suggest minor revision as:
- Usually, limitations of the study and further research are provided within the Conclusion section: please considered to move them from the Discussion section to the Conclusions one
- There are some typos, as page 2, line 67 or page 12, line 408 (double comma)
- Please consider using they/them/their instead of she or he/ her or him/ her or his (as page 2 line 62 or page lines 4 163 e 168)
Additionally, I suggest also to insert your work within the Sustainable Development Goals, as you are tackling a relevant topic for the framework.
Author Response
Dear Reviewer,
We agree with your first comment. However, another reviewer's contradictory comment was to shorten the Conclusions section. Thus, it was difficult to move the limitations, and we hope that this solution is acceptable.
Thank you for pointing out typos. Those have been corrected accordingly.
The comment regarding third persons was essential, thank you for noticing. We have corrected the language in this regard.
We acknowledge your suggestion about sustainable development goals. We included the following sentences to the manuscript:
“The United Nations has defined two relevant sustainable development goals (a) to ensure healthy lives and promote well-being for all at all ages, and (b) to promote sustainable, inclusive and sustainable economic growth, full and productive employment and decent work for all [6].” (rows 47 - 50)
"In line with the United Nations 'Sustainable Development Goals, the Ultimate aim is to better support employees' health and facilitate both an empathic workplace culture and Psychological safety in demanding knowledge-Intensive work." (row 493)
Sincerely,
The authors
Reviewer 2 Report
Dear Authors,
The problem raised in the manuscript is significant in the present times, when work is becoming more and more intense and workers are required to be more efficient. Like the authors, I am convinced that this is not insignificant for mental health.
I have no comments on the research methodology, results and their discussions.
However, doubts arise as to the size of the sample of employees that has been surveyed. The paper should provide data on the number of knowledge-intensive workers in Europe (population size). The correctness of the selection of the size of the tested sample should be justified or the analysis of the maximum error should be performed at the specified confidence level for the tested sample.
Author Response
Dear Reviewer,
We are grateful for the valuable comments we received.
Thank you for the useful advice on considering the number of European knowledge workers and confidence intervals. We did not find exact statistics on the number of knowledge workers, but we estimated it as follows:
According to European Commission, the European Union had 228.7 million employees in 2015. The current approximation is that 30 - 40 % of the European employees are knowledge-intensive workers. Thus, we used estimations of 70 and 100 million (knowledge workers) to calculate confidence intervals at 95 % confidence level. Both population sizes resulted in the same confidence intervals. To sum up, we reported the numbers (also in Figure 2 and Figure 3) with confidence intervals at 95 % confidence level. We hope you find this solution satisfactory.
Regarding structural equation modeling (SEM), the confidence intervals are not typically calculated or reported because the model is evaluated by the fit indices, AVE, CR, and Cronbach’s alpha values. These values indicated excellent reliability. In general, ten responses per estimated relationship is minimum, but in our model, this number was 181.
Sincerely,
The authors
Reviewer 3 Report
Dear authors,
I found the study very interesting and easy to read and to understand. I have some recommendations and questions. Kind regards.
I would remove the title 1.1. With the name "introduction" is enough.
The study aim should be before the methods section. Please remove it from line 58-84.
Did the online questionnaire establish any filter to check what kind or person was answering the questionnaire? Or how did you know that the person was a "knowledge worker".
I would move Table 1 and Table 2 and its related data to the beginning of the results sections.
Include a subtitle of "Statistical Analysis" before the information of the analysis.
The conclusion should be more concise. Some information it would be better to include it in the discussion.
Author Response
Dear Reviewer,
We are grateful for the valuable comments we received.
Regarding if the article is adequately referenced, we did not receive any explanation for your choice (must be improved), for example, on which topic we have insufficient references. Two other reviewers did not mention any weaknesses in our references. Thus, we did not revise the references yet. Nevertheless, we are willing to improve if you could give us more information, please. We hope you find this solution satisfactory.
We removed title 1.1 as suggested.
We believe that the study aim is quite often presented in the Introduction, and two other reviewers were ok with the current positioning of the study aim. However, we did not fully understand this suggestion, and we are willing to changes if you could provide us more details.
We did not implement any filtering this time. The filtering was based on respondents' own perceptions. With our consent, we specifically requested knowledge workers from the European organizations to respond. Moreover, we shared the invitation mainly via the distribution channels followed by knowledge-intensive organizations. We included this information in the manuscript (row 249). Thank you for asking; we will take this into account in further research.
Thank you for pointing out the inconsistency in the Methods section. We moved Table 1 (row 289) and Table 2 (row 301) at the beginning of the Results section.
We added the subtitle "Statistical analysis" in the Methods section as suggested (row 272).
Lastly, we compressed the Conclusions slightly (rows 469 - 471) and included some sentences (rows 392 - 395 and 437 - 439) in the Discussion as suggested. We hope that the current form of the Conclusions is acceptable.
Sincerely,
The authors